# Distribution of neurosensory progenitor pools during inner ear morphogenesis unveiled by cell lineage reconstruction

Sylvia Dyballa[1], Thierry Savy[2], Philipp Germann[3], Karol Mikula[4], Mariana Remesikova[4], Róbert Špir[4], Andrea Zecca[1], Nadine Peyriéras[2], Cristina Pujades[1]*

[1]Department of Experimental and Health Sciences, Universitat Pompeu Fabra, Barcelona, Spain; [2]Multilevel Dynamics in Morphogenesis Unit, USR3695 CNRS, Gif sur Yvette, France; [3]Systems Biology Unit, Center for Genomic Regulation, The Barcelona Institute of Science and Technology, Barcelona, Spain; [4]Department of Mathematics, Slovak University of Technology, Bratislava, Slovakia

**Abstract** Reconstructing the lineage of cells is central to understanding how the wide diversity of cell types develops. Here, we provide the neurosensory lineage reconstruction of a complex sensory organ, the inner ear, by imaging zebrafish embryos in vivo over an extended timespan, combining cell tracing and cell fate marker expression over time. We deliver the first dynamic map of early neuronal and sensory progenitor pools in the whole otic vesicle. It highlights the remodeling of the neuronal progenitor domain upon neuroblast delamination, and reveals that the order and place of neuroblasts' delamination from the otic epithelium prefigure their position within the SAG. Sensory and non-sensory domains harbor different proliferative activity contributing distinctly to the overall growth of the structure. Therefore, the otic vesicle case exemplifies a generic morphogenetic process where spatial and temporal cues regulate cell fate and functional organization of the rudiment of the definitive organ.

*For correspondence: cristina. pujades@upf.edu

## Introduction

A major challenge in developmental biology is to explain how spatiotemporally controlled cell specification and differentiation occur alongside morphogenesis in the construction of functional organs. The inner ear is an attractive model to take on this challenge since it is accessible to manipulation, and it undergoes dynamic self-organization. It contains a manageable number of distinct cell types, which develop rapidly on an organized schedule to generate the functional units of the mature organ – the sensory patches. The key cell types of the inner ear, the supporting cells and the hair cells of the sensory patches, and the sensory neurons that innervate them, originate early during embryonic development from progenitors located in the otic vesicle, a 3D-structure arising from the otic placode adjacent to the developing hindbrain (*Durruthy-Durruthy et al., 2014*; *Raft et al., 2007*; *Sapède et al., 2012*; *Satoh and Fekete, 2005*), and they are easy to score by morphology, position and specific markers (*Haddon and Lewis, 1996*; *Raft and Groves, 2015*; *Whitfield, 2015*; *Wu and Kelley, 2012*). All these features have established the inner ear as a model widely used for the study of patterning and cell specification (*Atkinson et al., 2015*; *Cai et al., 2015*; *Fritzsch et al., 2006a*, *2006b*; *Whitfield and Hammond, 2007*; *Wu and Kelley, 2012*).

Despite a good understanding about the molecular hierarchies, the knowledge of how individual progenitors behave throughout patterning, proliferation, and morphogenesis remains elusive. Neuronal vs. sensory specification is achieved through well-defined bHLH transcription factors: *atoh1* for

**eLife digest** Our ears, eyes and other sensory organs collect information about the world around us. In the inner ear – which is responsible for balance and hearing – specialized cells known as hair cells detect sounds and body position. This information is passed on to other cells called sensory neurons, which relay the information to the brain. Both of these cell types originate from a pool of cells known as "progenitors" located in a structure called the otic vesicle within the embryo, where the progenitor cells follow different sets of instructions to make hair cells or sensory neurons. Although we have identified many of the genes that are important for setting these instructions, it is not known how the progenitor cells behave in the inner ear or how they follow these guidelines.

Dyballa et al. used high-resolution imaging to reconstruct the histories of individual hair cells and sensory neurons in the inner ear of zebrafish embryos. The experiments combined techniques that allowed individual cells to be tracked over time and showed what types of cell they developed into. Dyballa et al. used these data to develop a map of the progenitor cells in the whole otic vesicle.

The findings help to fill the void in our understanding of the link between gene activity and tissue architecture in the inner ear. The next challenge is to use these findings as a basis to further explore models for how sensory neurons and hair cells develop, and to understand how to overcome the inner ear degenerati as we age.

hair cell formation (*Millimaki et al., 2007*; *Bermingham et al., 1999*), *neurog1* for sensory neuron determination (*Andermann et al., 2002*; *Ma et al., 1998*), and *neuroD* for sensory neuron differentiation and survival (*Jahan et al., 2010*; *Kim et al., 2001*). Signals arising in the surrounding tissues regionalize the otic vesicle along axes (*Maier and Whitfield, 2014*; *Radosevic et al., 2011*; *Riccomagno et al., 2002*, *2005*; *Sapède and Pujades, 2010*), and this multiple step process implies a gradual restriction of cell fates over time (*Whitfield and Hammond, 2007*; *Wu and Kelley, 2012*). However, the phenotypes of targeted mutants for these signaling pathways are not always easy to reconcile (*Raft and Groves, 2015*), due to the limited comprehension of how developmental gene regulatory networks are integrated. For this, cellular data are needed as it can address how patterns can be achieved while the cells proliferate and the tissue undergoes morphogenesis, which may affect cell positioning and exposure to signals, and therefore cell specification.

Recent developments in 4D-microscopy imaging and cell tracking tools permit now simultaneous measurements at high spatial-temporal coverage and resolution, and therefore the assessment of cell lineages and cell behaviors including displacements and proliferations (*Amat et al., 2014*; *Blanpain and Simons, 2013*; *Faure et al., 2016*; *Keller, 2013*; *Li et al., 2015*; *Olivier et al., 2010*; *Truong et al., 2011*). Thus, it is time to progress in filling the void between gene regulatory networks and tissue architecture. With this purpose, we reconstructed the otic neurosensory lineage and investigated their single cell behavior by using in vivo imaging technologies paired with image processing tools (*Figure 1*, *Figure 1—figure supplement 1*; *Faure et al., 2016*). Our dynamic analyses revealed some surprising results such as the impact of neuroblast delamination and otic vesicle morphogenesis on the size and shape of this progenitor domain, and further that place and order of neuroblast delamination foreshadow their position within the statoacoustic ganglion (SAG). The comparative map of neuronal and sensory progenitors in the whole otic vesicle allows understanding how their distribution changes over time, being largely segregated with a small region of putative overlap. These findings provide the cellular data helping to understand how gene regulatory networks may work during development, tissue degeneration and regeneration.

## Results

Genetic requirements for specification of otic neuroblasts are rather clear (*Andermann et al., 2002*; *Ma et al., 1998*); however, little is known about the cellular mechanisms underlying neuroblast development. Specifically we want to understand (i) how the neuroblast progenitor population is altered upon delamination of cells, (ii) how delamination coordinates in space and time, and (iii) how delaminated cells arrange to form the ganglion. We addressed these questions by exploring in depth and detail a selected number of embryos and support our findings with less extensive

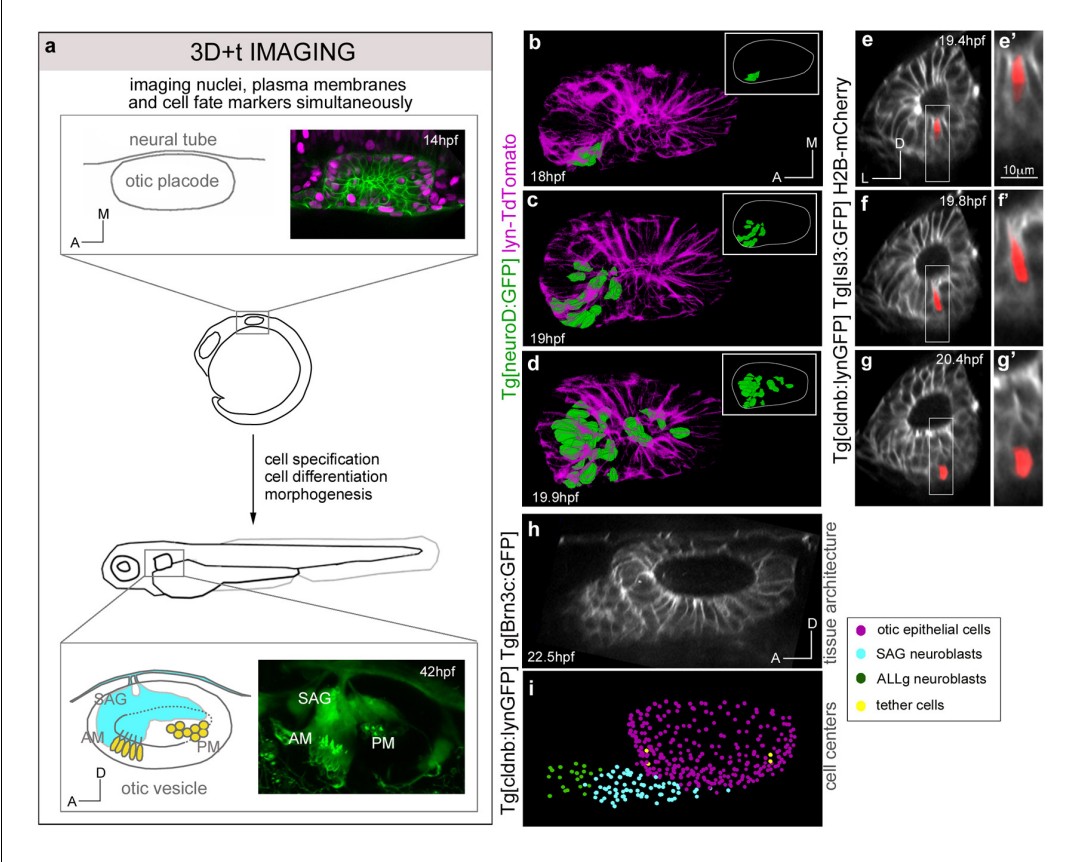

**Figure 1.** Expansion of the neuroblast delamination domain and formation of the SAG rudiment. (**a**) Overview of the imaging and image processing strategy: inner ears of zebrafish embryos stained for cell membrane, nucleus and cell fate markers were imaged between 14-42 hpf. Image datasets were processed by nucleus center detection, cell tracking and cell shape segmentation. Data were validated and curated (*Figure 1—figure supplement 1*). (**b–d**) Time-lapse stills showing the posterior expansion of the neuroblast delamination domain over time; 3D-rendering of segmented epithelial neuroblasts (green) in context of the otic structure (plasma membranes in magenta) at indicated times; insets display only the segmented delamination domain with the otic vesicle contour in white. ID Dataset: 140210aX; see *Figure 1—figure supplement 2d* for additional analyses. (**e–g**) Time-lapse stills showing a segmented delaminating neuroblast (red; *Video 2*); (**e'–g'**) magnifications of framed regions in (**e–g**). ID Dataset: 140426aX. (**h–i**) Still images from *Video 1* displaying: otic tissue architecture (**h**), and cellular distribution (**i**) upon SAG formation. Reconstructed cell centers are color-coded according to cell position/identity (see legend). ID Dataset: 140423aX. SAG/ALLg, statoacoustic/anterior lateral line ganglia. AM/PM, anterior/posterior maculae.

The following figure supplements are available for figure 1:

**Figure supplement 1.** 3D+time image analysis pipeline.

**Figure supplement 2.** Posterior expansion of the otic neuroblast delamination domain.

analyses of additional embryos, which we provide in supplementary form. Data from selected experiments can be downloaded from URL: http://bioemergences.eu/eLife2016.

## Expansion of the neuroblast delamination domain

In order to study the spatiotemporal dynamics of neuroblast delamination we in vivo imaged Tg[neuroD:GFP] embryos, which express GFP in neuronal progenitors just prior to and after delamination from the otic epithelium. We observed the first delamination events at 18 hpf in the most anterolateral region of the otic floor (*Figure 1—figure supplement 2a–a'*). This domain expanded towards middle and posterior regions with cells delaminating very close to the neural tube (*Figure 1—figure supplement 2b–c'*), as had been described by analysis of serial transverse sections from 22 hpf

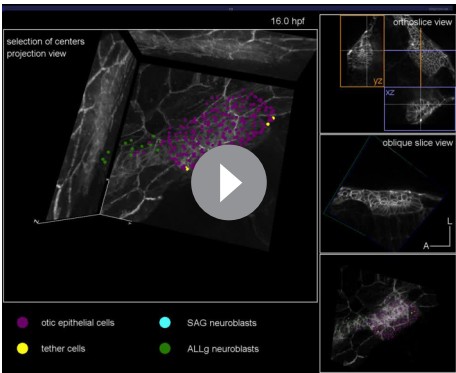

**Video 1.** Early organization of neuroblasts within the SAG. Tg[cldnb:lynGFP]Tg[Brn3c:GFP] embryos injected with H2B-mCherry mRNA were imaged, and reconstructed cell centers were color-coded according to their location/identity (see legend). The projection view video (large panel) simultaneously displays the topological organization of the cell group selection and tissue architecture as a projection of the GFP channel (plasma membranes in grey) in x,y,z-axes. The distinct visualization modes displayed on the right hand side allow for a detailed 3D-visualization of data during the analyses. Orthogonal views are used to validate cell tracking, the oblique slice view allows orienting the orthoplane along the embryonic axes, and the rendering view permits to display validated cell centers in the context of the whole image volume. ID Dataset: 140423aX.

axis by 22 hpf (*Figure 1h–i*; *Video 1*). To understand the origin of SAG-neuroblasts and how delamination coordinates in space and time we backtracked these neuroblasts to their progenitor state and followed their dynamics.

## Order and place of neuroblasts delamination from the otic epithelium prefigure their position within the SAG

As early as 20 hpf the SAG rudiment already innervates the different maculae, and by 48 hpf the SAG is composed of two segregated neuronal populations that although displaying the same molecular signature innervate different sensory patches (*Sapède and Pujades, 2010*; *Zecca et al., 2015*). We were eager to explore the spatiotemporal features controlling these neuronal populations and to gather information on population dynamics and lineage relationships. We first investigated whether neuroblasts were primed to different SAG-populations by backtracking SAG-neuroblasts to their progenitor state, and assessing individual cell position and tissue movements over time systematically. For

onwards (*Haddon and Lewis, 1996*). Segmentation of the epithelial neuroD-domain allowed us to illustrate the quick expansion of this territory from anterolateral to posteromedial (*Figure 1b–d*), and when comparing three different embryos this emerges as a common trend (*Figure 1—figure supplement 2d*). After the delamination domain is established many more cells delaminate from this territory, accumulating in the SAG just below the epithelium in close intimacy with the anterior lateral line ganglion (ALLg; *Figure 1i*, *Video 1*).

Neuroblast delamination implies that cells exit the otic epithelium basally and therefore undergo cell shape changes. Segmentation of individual cells -delineation of the cell contours by computational tools- when transiting from the epithelium to the SAG showed that the neuroblast cell body moves basally in less than one hour after undergoing apical constriction (see red cell in *Video 2*; *Figure 1e–g′*). Neighboring neuroblasts often delaminate consecutively to accumulate in the SAG, which quickly becomes an adjacent compact mass extending beneath the ventral floor of the otic vesicle along the anteroposterior

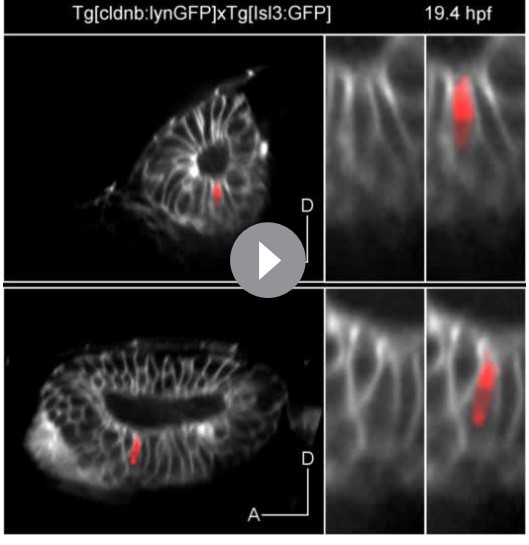

**Video 2.** Segmentation of delaminating neuroblasts. Tg[cldnb:lynGFP]Tg[Isl3:GFP] embryos were injected with H2B-mCherry mRNA at 1cell-stage, and single delaminating neuroblasts (n = 5) were automatically segmented. A representative segmentation (red colored cell) is shown. Transverse and lateral views (top and bottom rows) with their respective high magnifications on the right hand side. Note that the neuroblast changes shape and exits the otic epithelium basally within one hour. ID Dataset: 140426aX.

**Table 1.** Cohort of embryos and datasets used for the study.

Datasets used in this study with corresponding information about transgenic embryos and mRNA injections. Temporal frequency of image acquisition (timestep imaging) and corresponding imaging sequences are depicted. Each dataset corresponds to one imaged embryonic inner ear for the corresponding time period, except for dataset 140402aX in which both ears were imaged.

| ID dataset | Transgenic embryo | mRNA injection | Timestep imaging | Imaging sequences |
|---|---|---|---|---|
| 140210aX | Tg[neuroD:GFP] | lyn-TdTomato | 5 min | 14.5–31.5 hpf |
| 140125aX | Tg[neuroD:GFP] | lyn-TdTomato | 5 min | 16–37.5 hpf |
| 140306aX | Tg[neuroD:GFP] | lyn-TdTomato H2B-cerulean | 5 min | 12–32.9 hpf |
| 140426aX | Tg[cldnb:lynGFP]Tg[Isl3:GFP] | H2B-mCherry | 4 min | 18–36.2 hpf |
| 140430aX | Tg[cldnb:lynGFP]Tg[Brn3c:GFP] | H2B-mCherry | 4 min | 17–37.2 hpf |
| 140423aX | Tg[cldnb:lynGFP]Tg[Brn3c:GFP] | H2B-mCherry | 4 min | 16–26.5 hpf |
| 140507aX | Tg[Brn3c:GFP] | H2B-mCherry | 5 min | 24–43 hpf |
| 140326aX | Tg[Brn3c:GFP] | lyn-TdTomato | 10 min | 25–45 hpf |
| 140519aX | Tg[Brn3c:GFP] | H2B-mCherry MO-neurog1 | 5 min | 24–42 hpf |
| 140513aX | Tg[Brn3c:GFP] | H2B-mCherry MO-neurog1 | 5 min | 24–32.9 hpf |
| 140402aX | Tg[Brn3c:GFP] Tg[Isl3:GFP] | lyn-TdTomato H2B-cerulean | 10 min | 25–34 hpf |

this, we used the Tg[cldnb:lynGFP]Tg[Isl3:GFP] embryo injected with H2B-mCherry (ID Dataset: 140426aX, *Tables 1–2*).

Analysis of the dynamics of this process revealed that once the first delamination events are observed, delamination occurs massively. The lineage tree shows that within six hours a big bulk of neuroblasts delaminates (see white, yellow and orange lines in *Figure 2a* corresponding to single delaminated neuroblasts), and many neuroblasts can divide after exiting (see lines branching in *Figure 2a*) as previously described (*Vemaraju et al., 2012*). This lineage analysis allowed ascribing delamination time (see hpf in x-axis in *Figure 2a*) and delamination position to each neuroblast. Then, we color-coded the reconstructed cell centers according to these two criteria and displayed them (i) early when in the otic vesicle (*Figure 2b,d,f*), and (ii) late when within the SAG (c-c′,e-e′,g-g′). The first observation was that neuroblasts delaminated randomly in space and time (see inter-mingled colors of the reconstructed cell centers in *Figure 2b*), but their organization within the SAG relies on the delamination order as delaminated cells aggregate laterally to the preexisting ganglion (*Video 3*), thereby generating a mediolateral (ML) growth pattern with early-delaminated cells (white) located more medially than late-delaminated ones (red) (*Figure 2c–c′*; see *Figure 2—figure supplement 1a–b* for additional analysis). While the inner ear grows and undergoes morphogenesis, the SAG becomes squeezed in the space between the ventromedial wall of the ear and the neural tube, such that its organization is gradually converted from ML to dorsoventral (DV), with early-delaminated sensory neurons located in close contact with the neural tube (*Video 3*; *Figure 2c*). Thus, the time of neuroblast delamination foreshadows the ML gradient of neuronal differentiation in the SAG.

Then, we analyzed the epithelial origin of two functionally distinct SAG populations: anterior, mainly involved in vestibular, and posterior responsible for vestibular and acoustic functions (*Haddon and Lewis, 1996*). We found that the relative position of progenitors foresees their final location in the SAG: neuroblasts from the anterior portion of the SAG delaminated from the antero-lateral floor of the otic vesicle, while the ones of the posterior SAG derived from the posteromedial otic epithelium (*Video 4*). Next, we addressed whether the epithelial coordinates prefigure the position of neuroblasts in the SAG primordium. We observed that the anteroposterior (AP) coordinates of neuroblasts delamination (see differently colored epithelial cell center populations in *Figure 2d*) generally defined their relative AP position within the SAG (*Figure 2e–e′*; *Video 5*; see *Figure 2—figure supplement 1c–d′* for additional analysis). These data show that progenitors delaminating within similar spatial regions along the AP axis maintain similar relative positions later on, and indeed the analysis of their trajectories indicates that cells, while in the otic epithelium, maintain their neighbor relationships. No such correlation was observed regarding the ML axis, because cells that were

**Table 2.** Datasets used for the study of the different biological questions.

Datasets used for each addressed biological question, and Figures in which the corresponding analyzed data are displayed. Each dataset corresponds to one imaged embryonic inner ear, except for 140402aX in which both ears were imaged. All datasets correspond to control samples, except for 140519aX and 140513aX that correspond to MO-neurog1 embryos (see *Table 1*). Note that we have performed deep and detailed analyses in few datasets, and supported the observations and conclusions with partial analyses of other datasets mainly included in supplementary figures.

| Biological insight | ID dataset | Figures |
|---|---|---|
| Expansion delamination domain | 140210aX | *Figure 1b–d* |
| | | *Figure 1—figure supplement 2* |
| | 140125aX | *Figure 1—figure supplement 2d* |
| | 140306aX | *Figure 1—figure supplement 2d* |
| Segmentation delamination domain | 140210aX | *Figure 1b–d* |
| Segmentation delaminating neuroblast | 140426aX | *Figure 1e–g'* |
| | | *Video 2* |
| Otic vesicle architecture | 140423aX | *Figure 1h–i* |
| MovIT tools | 140423aX | *Video 1* |
| Delamination dynamics | 140426aX | *Figure 2a* |
| Neuroblasts lineage | 140426aX | *Figure 2b–g'* |
| | | *Videos 3–5* |
| | | *Figure 3—figure supplement 1* |
| | 140423aX | *Figure 2—figure supplement 1* |
| Clonal behavior of neuroblasts | 140426aX | *Figure 3a–c* |
| | | *Figure 3—figure supplement 1c* |
| Neuronal progenitor map | 140426aX | *Figure 3d* |
| | | *Video 8* |
| Spatiotemporal hair cell differentiation | 140507aX | *Figure 4a–c,e* |
| | | *Video 6* |
| | | *Figure 4—figure supplement 1a* |
| | 140326aX | *Figure 4—figure supplement 1c* |
| | 140402aX | *Figure 4—figure supplement 1b* |
| | 140519aX | *Figure 4f–h,j* |
| | | *Video 6* |
| | | *Figure 4—figure supplement 1a* |
| Hair cell progenitor map | 140507aX | *Figure 4d* |
| | | *Video 7* |
| | | *Video 8* |
| | 140326aX | *Figure 4—figure supplement 1c* |
| | 140519aX | *Figure 4i* |
| | | *Video 7* |
| Spatiotemporal cell proliferation | 140507aX | *Figure 5a–f* |
| | | *Video 9* |
| | 140519aX | *Figure 5g–l* |
| Local cell density/NN-distances | 140507aX | *Figure 5m* |
| | | *Video 10* |
| | | *Figure 5—figure supplement 1a,b* |
| | 140519aX | *Figure 5n* |
| | | *Video 10* |
| | | *Figure 5—figure supplement 1b* |
| | 140430aX | *Figure 5—figure supplement 1b* |
| | 140326aX | *Figure 5—figure supplement 1b* |
| Otic vesicle volume | 140426aX | *Figure 5—figure supplement 1e* |
| | 140507aX | *Figure 5—figure supplement 1c,e* |
| | 140430aX | *Figure 5—figure supplement 1e* |
| | 140513aX | *Figure 5—figure supplement 1e* |
| | 140519aX | *Figure 5—figure supplement 1d,e* |

originally separated in the epithelium were found close in the SAG (*Figure 2f–g'*; *Video 5*), suggesting that the AP and time cues prevail. These findings establish a link between the place and order of

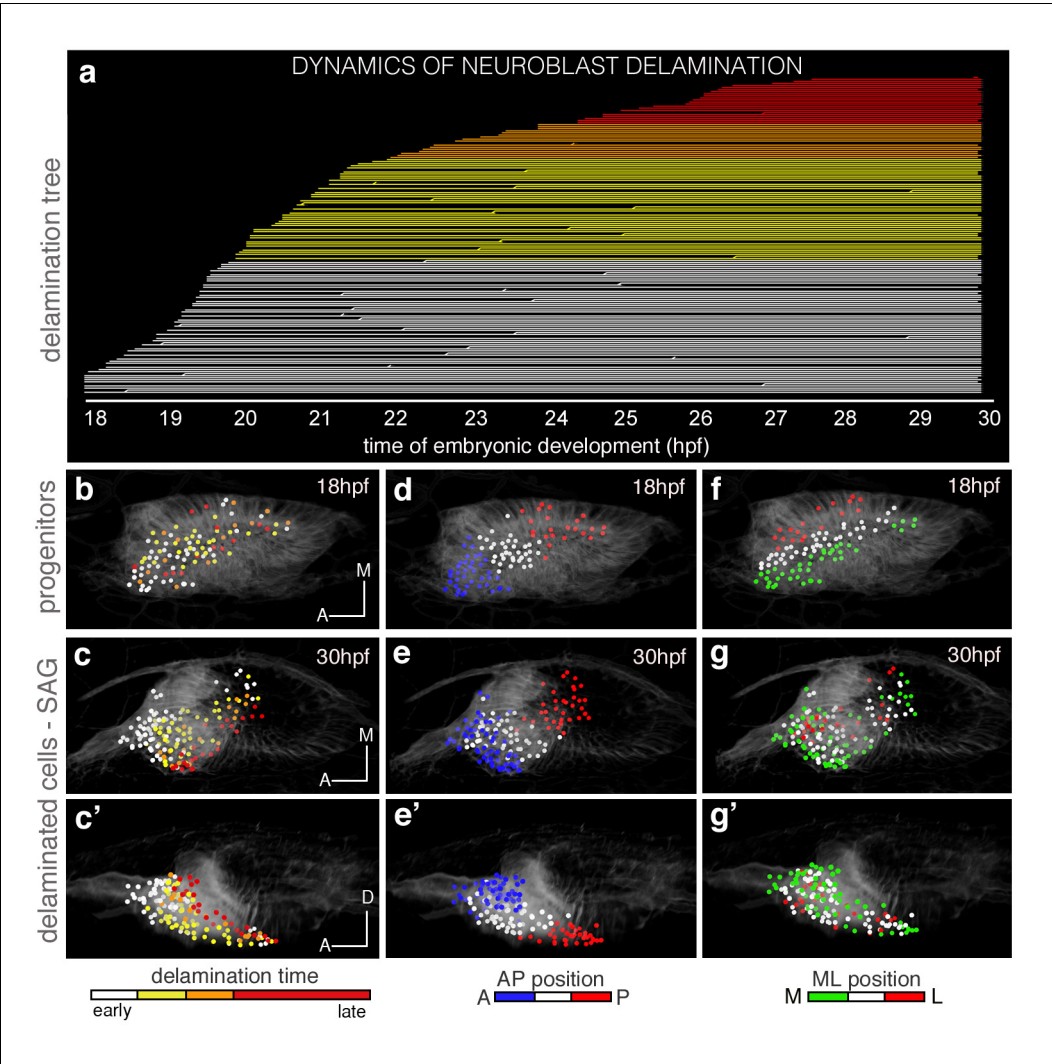

**Figure 2.** The organization of cells within the SAG relies on specific temporal and spatial cues. (**a**) Flat representation of the neuroblast lineage tree with branches indicating cell divisions. The x-axis displays the time of embryonic development in hours post-fertilization (hpf). Neuroblast lineages are displayed from the moment of delamination onwards and ordered and color-coded according to delamination timing (intervals: 18–20 hpf white, 20–22 hpf yellow, 22–24 hpf orange, 24–30 hpf red). Some cells were not tracked until the end of the sequence, and are depicted as interrupted lines. The extensive cell loss during the early stages of delamination (18–22 hpf) was verified in a second embryo; in both cases, about 25% (23.2% and 26.8%) of the otic epithelial cells at 18 hpf exit by delamination in the consecutive four hours. (**c–c',e–e',g–g'**) Neuroblasts within the SAG (n = 144 of roughly total n = 250) were backtracked to their progenitor state in the epithelium (n = 98; **b,d,f**; *Videos 3* and *5*). Cell lineages were color-coded for: time of delamination (**b–c'**; same intervals as in (**a**)), position in the epithelium along the AP (**d–e'**), or ML (**f–g'**) axes. Note that ML organization of neuroblasts within the SAG (**c–c'**) relies on their delamination order, and that the blue/white/red epithelial pattern (**d–e'**; neuroblasts AP position) but not the green/white/red one (**f–g'**; neuroblasts ML position) is maintained in the SAG over this time period (18–30 hpf). Reconstructed cell centers were displayed as colored-dots together with the corresponding raw images (plasma membranes in grey level). (**b,d,f**) dorsal views; (**c,e,g**) ventral views; (**c',e',g'**) lateral views. Anterior is always to the left. For this analysis, Tg[cldnb:lynGFP] Tg[Isl3:GFP] line was injected with H2B-mCherry mRNA at 1cell-stage (*Tables 1–2*). ID Dataset: 140426aX; see *Figure 2—figure supplement 1* for additional analysis.

The following figure supplement is available for figure 2:

**Figure supplement 1.** Time of delamination and position of epithelial neuroblasts prefigure their location within the SAG.

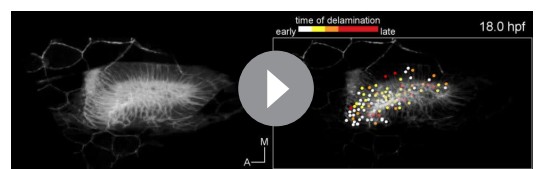

**Video 3.** The order of delamination foreshadows the mediolateral gradient of neuroblasts differentiation within the SAG. A cohort of 144 delaminated neuroblasts was analyzed for time of delamination. Reconstructed cell centers were color-coded according to four delamination intervals: 18–20 hpf white, 20–22 hpf yellow, 22–24 hpf orange, 24–30 hpf red. Note that neuroblasts exit randomly from the delamination domain. Those delaminating earlier are located more medially in the SAG than the later delaminating ones prefiguring the gradient of differentiation. Reconstructed cell centers were displayed as colored dots together with the corresponding volume rendering images (plasma membranes in grey level) with higher intensity on the left hand side. Tg[cldnb:lynGFP] Tg[Isl3:GFP] embryo was injected with H2B-mCherry mRNA at 1cell-stage. ID Dataset: 140426aX.

neuroblast delamination and their neuronal (functional) identity within the SAG, and suggest the existence of a spatial and temporal regulation in the otic epithelium of the fate and exit of SAG neuroblasts.

## Remodeling of the neuronal progenitor domain upon delamination

We were interested in understanding whether cell proliferation had any role in conferring neuroblasts categories. To investigate this question we used the previous dataset (ID Dataset: 140426aX, *Tables 1–2*) to undertake the neuroblast clonal analysis (*Figure 3a*), which showed that SAG-neuroblasts originate from a neuronal progenitor pool with different division behaviors: one third of the analyzed neuroblasts divided before exiting the epithelium (n = 42 red cells in *Figure 3a*, *Figure 3—figure supplement 1c*), one third divided after delamination (n = 40 orange cells in *Figure 3a*; *Figure 3—figure supplement 1c*), and a similar fraction of neuroblasts did not divide within this time interval (n = 34 blue cells in *Figure 3—figure supplement 1c*). Thus, cell position within the otic epithelium is not relevant for these cell division behaviors because no specific spatial distribution can be observed (*Figure 3—figure supplement 1c*). In addition, cell division and delamination are independent events, in contrast to what was proposed for epithelial invagination of the Drosophila tracheal placode (*Kondo and Hayashi, 2013*).

Neuroblasts dividing before delamination give rise to two daughter cells that will delaminate (*Figure 3a–c*). In most cases, sister cell delaminations happened within a two hours interval, however in a few cases the time of delamination between two sister cells can be as long as seven hours (*Figure 3b*). Neuroblasts giving rise to two daughter cells that delaminated within a delay interval of more than two hours were evenly distributed in the otic epithelium (n = 11/116 bicolored cell centers, *Figure 3—figure supplement 1d*). Overall, these analyses suggest that (i) epithelial neuroblasts' divisions give rise to cells that will delaminate (*Figure 3c*), and (ii) cell division is not necessary as a trigger for delamination.

Finally, we used the same dataset (ID Dataset: 140426aX; *Tables 1–2*) to understand how the neuronal progenitor domain changes upon cell delamination and morphogenesis of the inner ear over time (orange circles in *Figure 3d*; n = 131 at 18 hpf). We followed as well the behavior of the surrounding otic epithelial cells as a repair for the edge of the progenitor domain (see grey circles in *Figure 3d*, n = 64 at 26 hpf). Due to the massive cell delamination in a relatively short time (more than 150 delamination events in the period

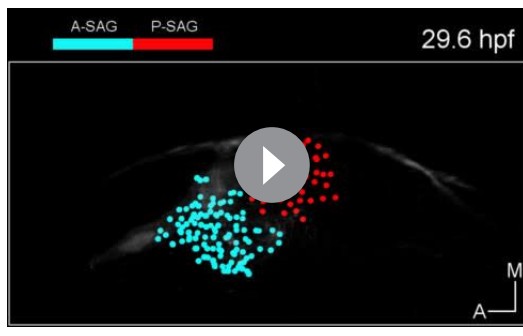

**Video 4.** The anterior and posterior SAG neuronal populations derive from the anterior and posterior otic epithelium, respectively. Neuroblasts belonging to distinct SAG neuronal populations, anterior or posterior (cyan/red), were backtracked to their otic epithelial origin prior to delamination. Note that cells of the anterior and posterior SAG delaminate from the anterior and posterior otic epithelium, respectively, and they maintain this relative position. The video is played backwards. These neuroblasts are the same set of cells as used in *Video 3*, but analyzed for AP position in the SAG instead of delamination time. ID Dataset: 140426aX.

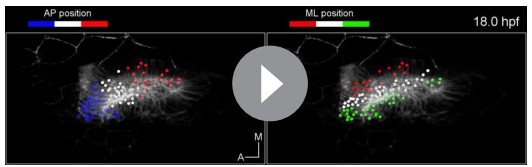

**Video 5.** Position of epithelial neuroblasts along the anteroposterior axis prefigures their location within the SAG. Reconstructed cell centers from neuronal progenitors were color-coded for position along the anteroposterior (AP) (left) or mediolateral (ML) (right) axes in the otic epithelium and followed from 18 hpf to 30 hpf. Note that the relative position of neuroblasts along the AP but not the ML axis is maintained from the otic epithelium to the SAG. These neuroblasts are the same set of cells as used in *Video 3*, but analyzed for epithelial position along the AP/ML axes instead of delamination time. ID Dataset: 140426aX.

18-26 hpf), the neuronal progenitor domain undergoes dramatic size and shape changes (*Figure 3d*, compare the territory containing orange cell circles at 18 hpf and 26 hpf), as described by *Haddon and Lewis (1996)*. Upon delamination a large amount of cells exits from the otic epithelium: about 25% of total otic cells at 18 hpf are lost by delamination between 18 hpf and 22 hpf. The exit of neuroblasts and the growth of the vesicle are part of the morphogenetic processes that progressively restricted the progenitor field to a more lateral region of the ventral floor of the vesicle (orange cell centers in *Figure 3d*, 26 hpf).

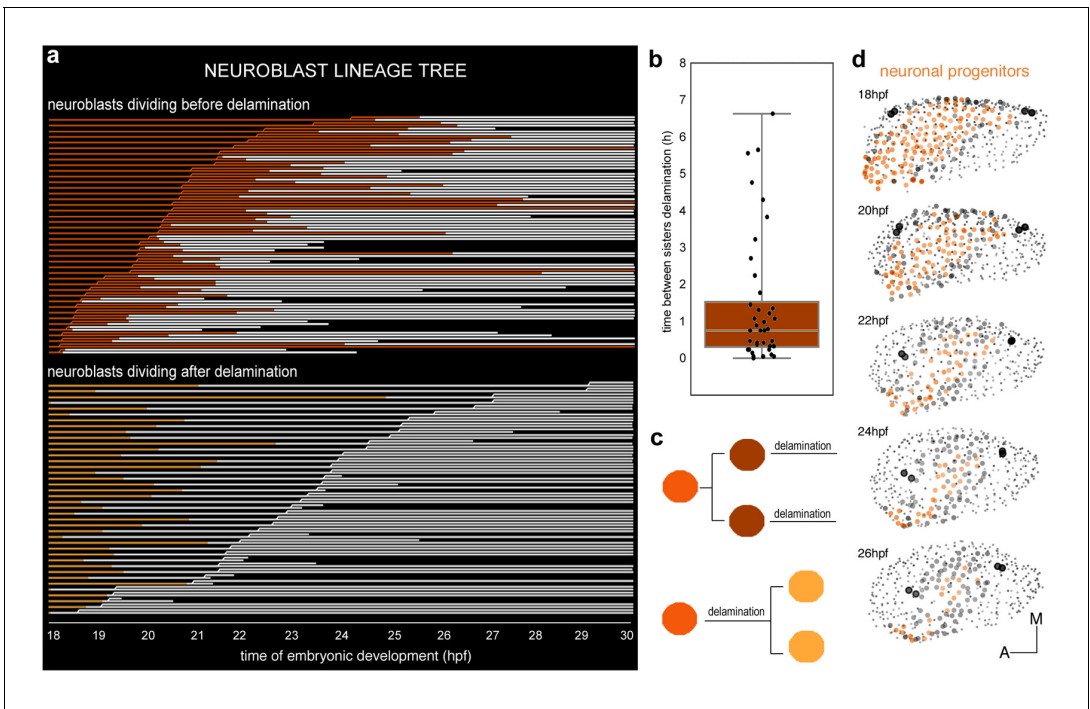

**Figure 3.** Clonal analysis of neuroblasts. (**a**) Lineages of neuroblasts (epithelial: colored; delaminated: white) ordered by time of division and grouped according to the division behavior -dividing before (red, top) or after (orange, bottom) delamination. Each line corresponds to a single neuroblast. Discontinued lines represent cells that were not tracked further. The x-axis displays the corresponding time of embryonic development (hpf). (**b**) Box plot illustrating the temporal delay in delamination between sister cells (*Figure 3—figure supplement 1d*). (**c**) Illustration of neuroblast division behavior colored as in (**a**). (**d**) Dynamic map of neuronal progenitors (orange circles) and their epithelial neighboring cells (grey circles) in the context of the whole otic vesicle (grey dots) over time; see *Video 8* for the 24 hpf animation. The color intensity of cell centers depicts the position of cells along the dorsoventral axis of the otic vesicle. The map was built after following the lineages from 18 hpf to 26 hpf of all encircled cells. Note how neuroblast delamination impacts on the size and position of the progenitor domain (orange cell centers) over time. Tether cells are depicted as black circles. For this analysis, Tg[cldnb:lynGFP] Tg[Isl3:GFP] line was injected with H2B-mCherry mRNA at 1cell-stage (*Tables 1–2*). ID Dataset: 140426aX.
The following figure supplement is available for figure 3:

**Figure supplement 1.** Spatial distribution of epithelial neuroblasts according to division behavior or delamination time.

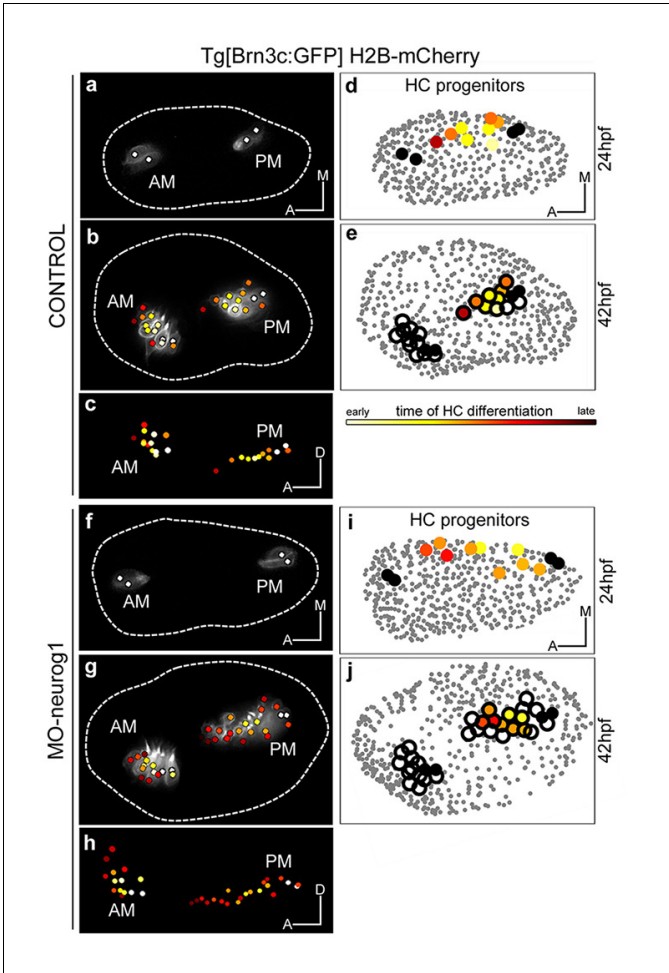

**Figure 4.** Spatiotemporal pattern of hair cell differentiation and map of sensory progenitors. Differentiated hair cells were tracked during 18 hr in control and MO-neurog1, and reconstructed cell centers were color-coded according to the differentiation time displayed in the legend (*Video 6*). (a–b,f–g) Spatiotemporal pattern of hair cell differentiation of the anterior/posterior maculae (AM/PM); reconstructed colored cell centers overlaid with the corresponding raw images (hair cell fate in grey level) from Tg[Brn3c:GFP] embryos; (c,h) reconstructed colored cell centers in lateral view. Note how the temporal but not the spatial development is altered in the MO-neurog1 PM (see *Figure 4—figure supplement 1*). (d,i) Map of hair cell progenitors in the whole otic vesicle (*Videos 7–8*); the maps were generated by backtracking the differentiated PM hair cells (e,j). ID Datasets: 140507aX for control, 140519aX for MO-neurog1; see *Figure 4—figure supplement 1* for additional analyses

The following figure supplement is available for figure 4:

**Figure supplement 1.** Temporal pattern of hair cell differentiation in AM and PM.

## Spatiotemporal development of the sensory patches

Hair cells are key cell types for building a functional sensory patch, and they are specified in the otic vesicle in the same way in all vertebrates. However, one difference is that in the fish the first hair cells differentiate while neuronal precursors are still delaminating, and that hair cells continue to be produced throughout life (*Haddon and Lewis, 1996*). This raises the question of how neuronal and hair cell production is coordinated from progenitor domains located very close within the otic epithelium, and this led us to compare the progenitor maps for hair cells and neurons.

Up to date, the formation of the anterior (AM) and posterior (PM) maculae (the first sensory patches to arise) has been followed up by gene or transgene expression mainly in 2D, and therefore their proper allocation within the 3D-otocyst was difficult to assess. In order to understand how

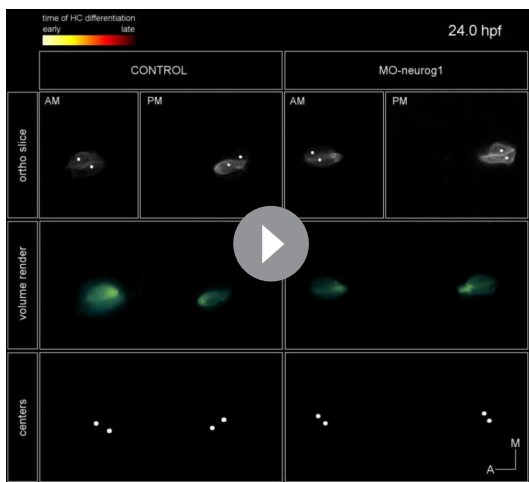

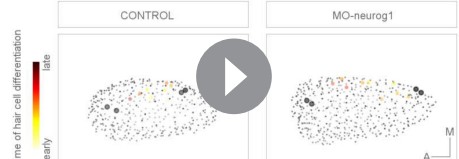

**Video 7.** Position of the posterior macula hair cell progenitor pools in control and MO-neurog1 embryos. Dynamic display of the posterior macula hair cell progenitors in control and MO-neurog1 embryos at 24 hpf. Progenitor pools were determined from the backtracking of differentiated hair cells in the Tg[Brn3c: GFP] line injected with H2B-mCherry at 1cell-stage. Hair cell progenitors are color-coded for time of differentiation and plotted in the context of the whole otic vesicle (grey dots depict reconstructed cell centers of the otic epithelium). Tether cells are shown as black circles. The animation displays a rotation of otic vesicles around the anteroposterior axis. ID Datasets: 140507aX for control, 140519aX for MO-neurog1.

**Video 6.** Spatiotemporal pattern of hair cell generation of sensory maculae in control and MO-neurog1 embryos. Tg[Brn3c:GFP] embryos were injected with H2B-mCherry mRNA at 1cell-stage (in the presence or absence of MO-neurog1), and differentiated hair cells of the anterior (AM) and posterior (PM) maculae were tracked during 18 hr (*Table 1*). The centers of hair cells were reconstructed and color-coded according to their time of differentiation as given by the onset of GFP expression. Top row displays reconstructed color-coded cell centers together with imaging data (orthoslice views of the maculae with raw images of hair cells in grey level), middle row shows imaging data alone (GFP signal as volume rendering), and bottom row displays only the reconstructed color-coded cell centers. Volume rendering and reconstructed cell centers panels rotate from dorsal to lateral view to illustrate the 3D-organization of hair cells within the maculae. ID Datasets: 140507aX for control, 140519aX for MO-neurog1.

these sensory patches arose we followed in vivo the incorporation of newly differentiated hair cells. The first hair cells to arise -the so called tether cells (*Riley et al., 1997*)- constituted the posterior pole of each of the maculae (white-dotted cells in *Figure 4a–c*; *Figure 4—figure supplement 1b,c*) and prefigured the position of these sensory patches within the otic vesicle (*Video 6*). Next, these two sensory patches gradually increased their size incorporating differentiated hair cells at their anterior pole with a specific pattern (see color-coded cell centers in *Figure 4b–c*; *Video 6*; ID Dataset: 140507aX; and see *Figure 4—figure supplement 1b,c* for additional samples exhibiting the same pattern). At the same time, growth and morphogenesis of the vesicle led to a structure in which the AM remained anterior and ventral, and the PM positioned in the posterior ventral edge of the medial wall of the vesicle. Once hair cells differentiate they become postmitotic (*Video 6*). Thus, the formation of the sensory patches depends on a pool of progenitors providing postmitotic differentiated hair cells. AM and PM develop asynchronously, with the incorporation of new hair cells being delayed in the PM (*Figure 4—figure supplement 1*; *Sapède and Pujades, 2010*).

Previous findings indicated that a specific pool of neuronal progenitors switches its fate to hair cells of the PM upon abrogation of neurog1 function (*Sapède et al., 2012*). However, they were unable to clearly assess the position of this progenitor pool and how it behaved. To do so, we first undertook the dynamical analysis of maculae generation upon neurog1-downregulation by injection of translation-blocking morpholino (ID Dataset: 140519aX, *Tables 1–2*), which fully recapitulates the phenotype of neurog1[hi1059] mutants (*Sapède et al., 2012*). In the absence of neurog1, sensory neurons do not form and supernumerary hair cells are produced in the PM (*Figure 4f–h*; *Video 6*). Tracking the differentiated hair cells allowed us to compare the dynamics of sensory development at the single-cell level: the overall spatial pattern of hair cell generation in the AM/PM was very similar between control and morphant embryos (compare *Figure 4b–c with g–h*); however the kinetics of hair cell production in the PM differed due to a boost of differentiation in MO-neurog1 from 34 hpf onwards (*Figure 4—figure supplement 1a*). All supernumerary hair cells in the MO-neurog1

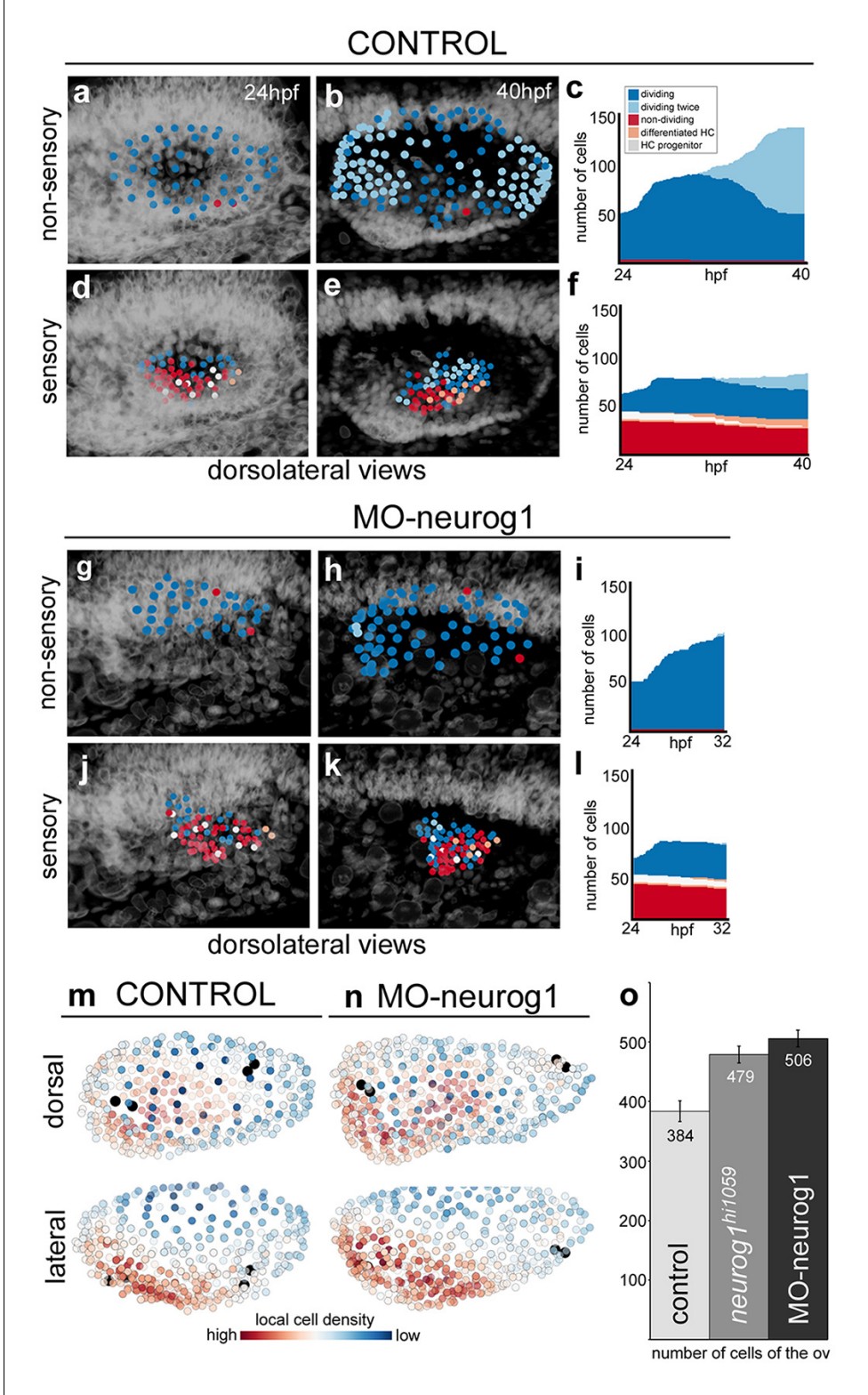

**Figure 5.** Heterogeneous cell behavior in the non-sensory and sensory domains. Neighboring cells in the non-sensory and sensory domains of control (a–f) and MO-neurog1 (g–l) were tracked and reconstructed cell centers were color-coded according to cell proliferation/differentiation status (see legend in (c); *Video 9*); they were plotted on the top of the corresponding raw images (a–b,d–e,g–h,j–k; nuclei in grey level), or in graphs over time (c,f, i,l) displaying the total number of cells in each domain and their status in the course of the video. Note the differences in the graphs between non-sensory and sensory domains, but not between control and MO-neurog1 embryos. (m–n) Estimated local cell densities at 24 hpf are represented by

*Figure 5 continued on next page*

*Figure 5 continued*

color-coded cell centers across the whole otic epithelium (*Video 10*). Tg[Brn3c:GFP] embryos injected with H2B-mCherry and with/without MO-neurog1 at 1cell-stage were used for full lineage reconstruction (*Tables 1–2*). Anterior is always to the left. ID Datasets: 140507aX for control, 140519aX for MO-neurog1; see *Figure 5—figure supplement 1* for additional analyses. (o) Graphic depicting the total number of cells in the otic vesicles for wild type (control, n = 3), *neurog1^{hi1059/ hi1059}* mutant in the Tg[Isl3:GFP] background (n = 3), and MO-neurog1 embryos (n = 2) at 24 hpf.

The following figure supplement is available for figure 5:

**Figure supplement 1.** Tissue architecture in sensory and non-sensory domains.

originate from newly differentiated cells, and not from differentiated hair cells re-entering the cell cycle (*Video 6*).

To unveil the organization of the PM hair cell progenitor pool we generated the map of hair cell progenitors in the whole otic vesicle over time (ID Datasets: 140507aX, 140326aX). For this, differentiated hair cells at 42 hpf were backtracked to their progenitor state, and information about differentiation time (color-code) and progenitor distribution was combined (*Figure 4d–e*; *Figure 4—figure supplement 1c*). 3D-reconstructions revealed that progenitors for hair cells of the PM were distributed over the ventromedial domain of the otic vesicle at early stages of embryonic development (see color-coded cell centers in *Figure 4d* and *Figure 4—figure supplement 1c*), and that upon neurog1 inhibition the hair cell progenitor pool expanded along the anteroposterior dimension allocating more medially (ID Dataset: 140519aX; *Figure 4i*, *Video 7*). Furthermore, the position of hair cell progenitors foreshadows the organization of differentiated cells within the sensory patch, demonstrating that otic epithelial cells did not rearrange during early stages of hair cell differentiation. However, the relative positions of the maculae changed upon morphogenesis, resulting in the growth of the PM towards the anterior region while growth of the AM was mainly anterior and towards dorsal (*Figure 4c,h*). Finally, our analysis allowed us to compare the progenitor maps for hair cells and neurons at the onset of hair cell differentiation (24 hpf): they were largely segregated with a small region of putative overlap (*Video 8*), consistent with the existence of a pool of dual progenitors (*Sapède et al., 2012*).

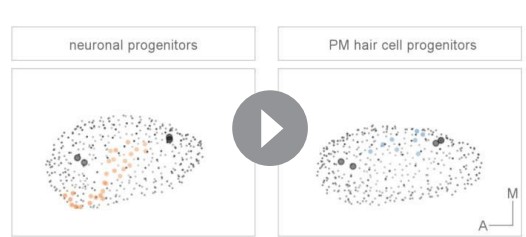

**Video 8.** Maps of neuroblasts and PM hair cell progenitors in the whole otic vesicle. The neuronal progenitors (orange; *Figure 3d*) and the posterior macula hair cell progenitors (blue; *Figure 4d*) are plotted in the context of the whole otic vesicle (grey dots) at 24 hpf. Tether cells are shown as black circles. The animation displays a rotation of otic vesicles around the anteroposterior axis. Note that the two progenitor domains are adjacent, and neuroblasts are located more ventrally while sensory progenitors are more medially. ID Datasets: 140426aX for neuronal progenitors, 140507aX for PM hair cell progenitors.

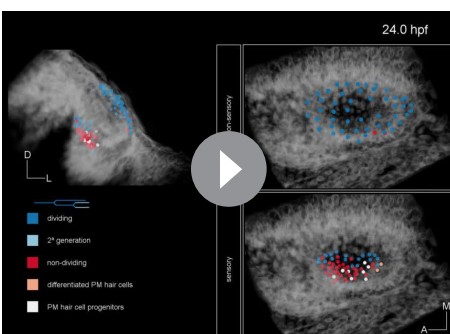

**Video 9.** Cell proliferative activity in the non-sensory and sensory domains. Tg[Brn3c:GFP] embryos injected with H2B-mCherry at 1cell-stage were used for full lineage reconstruction (from 24 hpf to 38 hpf) of 51 and 64 neighboring cells located in the non-sensory and sensory domains of the otic vesicle, respectively. Transverse view on the left is to better illustrate the position of the cell population domains along the axes. Reconstructed cell centers color-coded according to cell division/differentiation status and plotted in context of the raw images (nuclei channel as volume rendering in grey levels). ID Dataset: 140507aX.

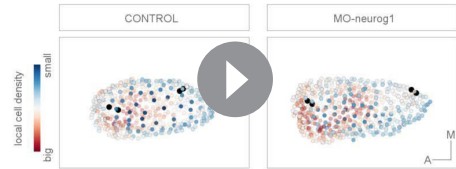

**Video 10.** Local cell densities across the whole epithelium in control and MO-neurog1. All cells of the otic epithelium in a control and a MO-neurog1 embryo are plotted and color-coded according to their estimated local cell density at 24 hpf. Tether cells are depicted in black for better orientation. The animation displays a rotation of otic vesicles around the anteroposterior axis. Note that the anteroventral territory displays higher cell density in both cases. ID Datasets: 140507aX for control, 140519aX for MO-neurog1.

## Sensory and non-sensory domains of the otic vesicle display different proliferative activity

As overall shape and size of the otic vesicle remain relatively constant during neuroblast delamination, we assumed that there was some homeostatic process involving cell proliferation elsewhere and we further validated this hypothesis.

To understand the proliferative behavior of the distinct otic territories, we reconstructed during 16 hr the lineages of 51–64 neighboring cells located either in the non-sensory (dorsolateral region of the otic vesicle; *Figure 5a–c*, *Video 9*) or in the sensory domain (ventromedial epithelial region; *Figure 5d–f*, *Video 9*). For this we used the Tg[Brn3c:GFP] H2B-mCherry injected embryo (ID Dataset: 140507aX). Cell behavior was assessed according to cell division (dividing/non-dividing) or cell differentiation (progenitor/differentiated) status. Cell position was monitored over time to identify spatial organization related to these features (*Video 9*). Cells within the non-sensory domain actively proliferated (*Figure 5a–c*; *Video 9*); in contrast, half of the analyzed cells in the ventral region did not divide and some differentiated into hair cells (*Figure 5d–f*; *Video 9*), suggesting that once hair cell progenitors are committed they divide less. Interestingly, even though the non-sensory domain displayed higher proliferative activity, it was less compacted (note neighboring reconstructed cell centers are more spaced in *Figure 5a–b* than in d–e), supporting the idea that it contributes to the overall growth of the vesicle during this time window. To quantify this, we calculated the nearest neighbor distance (NN-distance) for all cells in these domains over time and plotted the reconstructed cell centers color-coded for their NN-distance in the whole vesicle (see that green cell centers are more compacted than blue ones in *Figure 5—figure supplement 1a*; see *Figure 5—figure supplement 1b* for analyses of additional specimens).

Similar cell behaviors and cell organization were observed in the MO-neurog1 with no increase in proliferative activity in the sensory domain (*Figure 5g–l*), revealing that upon neurog1 abrogation hair cell progenitors do not divide more actively, and therefore supernumerary hair cells derive either from expansion of the progenitor pool, or from precocious differentiation of progenitors.

Since in MO-neurog1 cells do not undergo delamination, we determined how these non-delaminated cells located within the epithelium. For this, we estimated the distribution of local cell densities over the entire otic epithelium (*Figure 5m–n*; *Video 10*). While upon neurog1 abrogation the number of cells was greatly increased (*Figure 5o*; control = 384 ± 17 cells n = 3 vs. *neurog1*$^{hi5109/hi5109}$ = 479 ± 14 cells n = 3 or MO-neurog1 = 506 ± 14 cells n = 2), the global pattern of cell densities did not change: cells in either condition are most densely packed in the anterolateral and ventral region and most spaced in the dorsal domain (see red and blue cells in *Figure 5m–n*; *Video 10*). This suggests that the impact of non-delaminated cells in the compaction of the otic epithelium is low. Thus, if in the MO-neurog1 cell density is not altered but cell numbers are higher in the vesicle, the volume of the otic structure should be larger than the control one. Indeed, when assessing the volumes by 3D-point-cloud segmentation the otic vesicles of MO-neurog1 were larger than the control ones (*Figure 5—figure supplement 1c–e*). Altogether these results show that the otic epithelium is a heterogeneous tissue, where proliferative activity and cell compaction differ between sensory and non-sensory domains during the generation of the neurosensory cellular elements.

## Discussion

We provide information about cellular/population dynamics and lineage relationships of neurosensory elements in the inner ear from the reconstruction of their lineage trees from 4D in vivo data. These results enable us to: (i) understand the proportions of the system, (ii) reveal the impact of morphogenesis in the spatiotemporal distribution of neurosensory cell progenitor pools, and (iii) provide

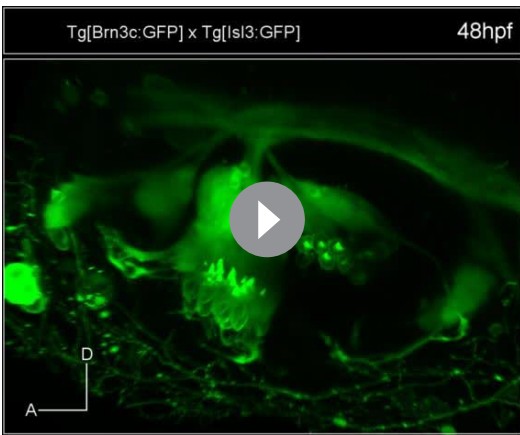

**Video 11.** Innervated sensory patches in the embryonic inner ear. Animation of a Tg[Brn3:GFP]Tg[Isl3:GFP] embryo displaying the sensory patches with differentiated cellular neurosensory elements in green at 48 hpf. Differentiated hair cells of the maculae (AM/PM) and cristae (ac/lc/pc) are innervated by sensory neurons of the SAG, which shows the typical segregation into anterior and posterior portion alongside with the segregated projections to the hindbrain (*Sapède and Pujades, 2010*).

the cellular data to complement the well described gene regulatory networks involved in neurosensory specification. Our strategy has been to analyze in depth and detail few selected embryos, and support our findings with lower power analyses of other specimens. This analysis of progenitor populations in their native environment over extended timespans provides the opportunity to understand in vivo cell behaviors, both at the single cell level and at the cell population level. However, due to the technically demanding nature of the experiments and analyses the sample size could not be very high. All biological systems have intrinsic noise, and therefore variability: absolute cell numbers may differ between embryos, the onset of developmental processes might occur at slightly different times, or even distinct transgenic backgrounds may behave different. Therefore, to overcome these possible biases we analyzed additional specimens with lower power analysis and demonstrate that our approach can yield reproducible results in terms of cell type proportions and cellular behaviors.

To assure the robust outcome of a functional inner ear, with sensory patches containing the precise number of mechanosensory hair cells properly innervated by sensory neurons arranged in a precise topology (*Video 11*), the developmental strategy used by distinct progenitor populations differs: neuronal specification is concomitant with proliferation (before/after delamination), while hair cell specification and differentiation lead to postmitotic cells indicating that the final number of sensory cells relies on the control of the progenitor pool. Interestingly, the position of hair cell progenitors foreshadows the organization of differentiated cells within the sensory patch, demonstrating that although the otic vesicle undergoes morphogenesis and cells do extensively proliferate, progenitors do not rearrange within the epithelium during early stages of hair cell differentiation.

The inner ear comprises two distinct functional modalities –vestibular and acoustic- carried out by different SAG populations. However, their epithelial origin and how SAG neuroblasts acquire their specific identity are still open questions. Here we show the importance of place and time of neuroblast delamination in their allocation within the SAG, and shed light on how distinct information may converge in the progenitor cells. Delamination place confers position along the AP axis of the SAG, and therefore function, most probably as the result of the integration of patterning signals involved in the emergence of the different domains (*Maier and Whitfield, 2014*; *Radosevic et al., 2011*). Additionally, delamination time prefigures the relative position of early-delaminated neuroblasts vs. late-delaminated ones and therefore, the gradient of neuronal differentiation within the SAG. Consequently, an epithelial neuroblast will need to integrate spatial (its position in the epithelium) and temporal (its time of delamination) information for its final allocation within the SAG. However, to achieve the fully functional organ, complexity needs to increase later on possibly by some new delamination events, division of SAG neuroblasts and changes in morphology. Given that auditory neurons, which innervate the PM, accumulate later than the vestibular neurons (*Sapède and Pujades, 2010*), it is possible that the majority of neuroblasts becoming auditory neurons may delaminate from posterior domains, since no specific increase in cell proliferation has been observed at later stages in the posterior part of the SAG (*Vemaraju et al., 2012*). However, if this is the mechanism and how it is controlled remain to be explored.

The knowledge of how the gradient of neuronal differentiation within the SAG is built up helps the comprehension of its differential gene expression pattern according to the neuronal differentiation state (*Zecca et al., 2015*), and the organized distribution of differentiated neurons

(*Vemaraju et al., 2012*). Thus, our results provide (i) a model to discuss how the selection of different progenitors and the determination of their relative population size could be regulated, which would not be possible to derive by gene expression only, and (ii) a framework to challenge the system –both in silico and in vivo- upon modification of key factors for cell fate decisions and patterning.

Interesting to note is that during a relatively short period of time, 18-30 hpf, a big proportion of epithelial neuroblasts delaminate. This drives a dramatic remodeling of the neurosensory progenitor domain that may lead to changes in the exposure of progenitor domains to sources of signals, such as Shh from the floor plate and the ventral neural tube. This loss of epithelial cells is probably compensated for by the active proliferation in the dorsal non-sensory territory in order to keep the homeostasis of the organ.

Finally, we provide the dynamic map of neurosensory progenitors based on in vivo cell lineage studies supplying a global and temporal perspective to previous otic neurosensory lineage analyses, which were mainly focused on the spatial dynamics of gene expression (*Durruthy-Durruthy et al., 2014*). This permits targeting and challenging progenitor pools specifically. Therefore, our findings establish a ground to further explore intrinsic vs. extrinsic models for cell fate determination, and will contribute to the mechanistic understanding of the developmental gene regulatory networks.

## Materials and methods

### Zebrafish strains

Zebrafish embryos were obtained by mating of adult fish using standard methods. All fish strains were maintained individually as inbred lines. All protocols used have been approved by the Institutional Animal Care and Use Ethic Committee (PRBB–IACUEC), and implemented according to national and European regulations. All experiments were carried out in accordance with the principles of the 3Rs. Wild type zebrafish strain was AB/Tu (RRID:ZIRC_ZL1/RRID:ZIRC_ZL57). Tg[neuroD: GFP] expresses GFP in neuronal progenitors (*Obholzer et al., 2008*) and differentiating neuroblasts (*Zecca et al., 2015*), and Tg[Isl3:GFP] (also called Isl2b) expresses GFP in afferent sensory neurons of cranial ganglia (*Pittman et al., 2008*). Tg[cldnb:lynGFP] labels the plasma membranes of the otic and anterior lateral line structures (*Haas and Gilmour, 2006*) and allows to visualize the kinocilium of differentiated hair cells. Tg[Brn3c:GFP] expresses GFP in differentiated hair cells of the ear and lateral line system, coinciding with the onset of differentiation (*Xiao et al., 2005*). Embryos homozygous for the *neurog1$^{hi1059}$* mutation (*Golling et al., 2002*) in the Tg[Isl3:GFP] background were obtained by incross of heterozygous carriers (*Sapède et al., 2012*); the presence of the *neurog1$^{hi1059}$* allele was identified by PCR genotyping fin-clips or embryo tails genomic DNA.

### Phalloidin and DAPI staining for assessing the total number of cells in otic vesicles

Tg[*neurog1$^{hi1059/+}$* Isl3:GFP] fish were crossed and embryos fixed overnight at 4°C in 4% Paraformaldehyde (PFA) and washed with 0.1% Tween-20 in PBS (PBST) at room temperature (RT). To label the plasma membranes, embryos were incubated for 12 hr at RT with Phalloidin-Alexa$^{658}$ (Invitrogen, Carlsbad, CA) diluted 1:20 in PBST containing 1.5% Triton X-100. They were washed with PBST and incubated with DAPI to stain nuclei. Embryos were then used for the analysis of the total number of cells within the otic vesicle (*Figure 5o*).

### Antisense morpholinos and expression constructs

Embryos were injected at 1cell-stage with translation-blocking morpholino oligomers (MOs) obtained from GeneTools, Inc.: neurog1-MO 5' -ACG ATC TCC ATT GTT GAT AAC CTG G-3' (*Cornell and Eisen, 2002*). 5 ng of MO-neurog1 was injected as previously described (*Sapède et al., 2012*). Morphants display the same phenotype as *neurog1$^{hi1059/hi1059}$* mutants (*Sapède et al., 2012*). For in vivo imaging morphant embryos were used for practical reasons: it is not possible to phenotype the mutants as early as the stage we start the imaging (24 hpf), and since *neurog1$^{hi5109/hi5109}$* need to be obtained by incross of *neurog1$^{hi5109/+}$* fish because the line cannot be maintained in homozygosis, only 25% of embryos are *neurog1$^{hi5109/hi5109}$* with the inconveniences that this poses for life imaging.

For mRNA expression, capped *H2B-mcherry*, *H2B-cerulean* (*Olivier et al., 2010*) or *lyn-TdTomato* mRNAs were synthetized with mMessage mMachine kit (Ambion). Embryos were injected at 1cell-stage and let to develop until the desired stages.

## Confocal imaging of stained whole mount samples

Stained fixed samples were mounted in 1% Low Melting Point (LMP)-agarose on glass-bottom Petri dishes (Mattek) and imaged on a Leica SP5 inverted confocal microscope using a 20x objective (NA 0.7).

## Embryo mounting and SPIM imaging

Embryos were anesthetized in 0.17 mg/ml tricaine in system water and mounted in 0.75% LMP-agarose in glass capillaries size 2 (volume 20 µl, BRAND GMBH). Time-lapse imaging was performed at 26.5°C (to avoid melting of LMP-agarose) on a Zeiss Lightsheet Z.1 microscope using a 20x or a 40x objective and the developmental stage was corrected accordingly (at 26.5°C development is delayed about 0.7 fold). Nuclei, plasma membranes and cell fate were recorded simultaneously for the entire system (*Figure 1a*, *Figure 1—figure supplement 1*). The cohort of embryos and datasets used in this study are depicted in *Tables 1* and *2*. Each dataset corresponds to the imaging of a distinct embryo inner ear, except for 140402aX in which both ears were imaged. Note that we have performed deep and detailed analyses in few datasets, and supported the observations and conclusions with partial analyses of additional datasets mainly included in supplementary figures.

## 3D+time image analysis pipeline

### Image pre-processing (*Figure 1—figure supplement 1*)

Image pre-processing was done using the Zeiss ZEN software and involved dual (illumination) side fusion and deconvolution (Regularized Inverse Method). To compensate for morphogenetic movements during image acquisition a semi-automated rigid registration was carried out using developed FIJI-scripts: the user is guided through the time steps of the 16-bit. czi data set and chooses by clicking a fixpoint of the structure (eg. a tether cell or a nucleus on the dorsolateral wall of the otic vesicle) for which the x, y, z coordinates are recorded. After navigating through all time steps of the video and having specified the fixpoints for all time steps the dataset is processed according to the recorded coordinates. This registration allows as well decreasing the size of region of interest, which substantially reduces data size. A separate 8-bit. vtk file with the corresponding transformation is generated for each time step and channel. Changes on the levels of the transgene fluorescent protein expression were compensated for by modifying dynamic ranges upon 16-bit to 8-bit mapping. Different datasets were generated for this work as displayed in *Tables 1–2*.

### Center detection and automatic tracking (*Figure 1—figure supplement 1*)

The .vtk files were then uploaded to the Bioemergences platform (*Faure et al., 2016*) and center detection was launched. Detected centers were validated using the CenterSelect application available in the platform. Subsequently, automatic tracking of the validated centers was launched and once completed the tracking data was validated and curated using MovIT, a custom-made graphical interface that offers the tools for segmentation and tracking of cells to accurately reconstruct their positions, movements and divisions (*Faure et al., 2016*). Videos of the developmental processes displayed in the manuscript were generated using FIJI (RRID:SCR_002285), MovIT, or a combination of both. Only validated/curated cell tracks were used for further analyses. MovIT has several display modes of time-lapse data such as: (i) orthoslices through the volume showing one xy-plane, one xz-plane and one yz-plane, used to validate cell tracks, (ii) oblique volume slices, which show a slice of a density map that is not parallel to the faces of the volume box and allows orienting the image along the axes, and (iii) volume rendering allowing to display a 2D projection of a 3D discretely sampled dataset; this permits to display cell centers in the context of the raw images in the whole volume (*Video 1*).

### Datasets analyses (*Figure 1—figure supplement 1*, *Tables 1–2*)

All used datasets are clearly stated in the Figure legends, and mostly throughout the text. For the comparison of the posterior expansion of the delamination domain in several specimens, the relative

position of delamination events along the AP axis of the otic vesicle was analyzed: the position of posterior-most neuroD:GFP expressing cells in the otic epithelium, and the position of the anterior and the posterior edges of the otic vesicle (dorsal view) were recorded over time (*Figure 1—figure supplement 2d*). Then the posterior edges of the delamination domains were plotted in percent of the AP otic vesicle length as a function of time.

For manual segmentation of the delamination domain, ITK-Snap software was used on three time steps of an imaging sequence of Tg[neuroD:GFP] embryos injected with lyn-TdTomato mRNA (*Figure 1b–d*). The delamination domain (identified by otic epithelial neuroD:GFP expression) and the otic vesicle volume (determined from the lyn-TdTomato signal) were segmented, and the resulting. vtk files were used to display neuroD-epithelial neuroblasts and membrane signal of the otic epithelium only (FIJI-3D viewer).

For cell lineage analysis, validated centers were grouped in selections and displayed with different appearances either alone or in context of the original data (*Figure 1h–i*; *Figure 2b–g'*; *Figure 2—figure supplement 1b–d'*; *Figure 3—figure supplement 1*; *Figure 4a–c,f–h*; *Figure 4—figure supplement 1b–c*; *Figure 5a–b,d–e,g–h,j–k*; *Figure 5—figure supplement 1a*). Cell lineages were also displayed as lineage trees (*Figure 2a*, *Figure 3a*, *Figure 2—figure supplement 1a*).

For analysis of cell proliferative activities over time, cell centers were grouped in selections according to cell states (eg. dividing or non dividing) and these data were further explored within SciPy, a Python-based ecosystem for scientific computing (*Figure 5*). Namely the data was imported as Pandas DataFrames; different time points and samples were registered using the iterative closest points transformation algorithm as implemented in. vtk and the results were plotted using matplotlib and seaborn.

To estimate the local cell density the other cell centers within 20 µm of each center were counted, and this number was then divided by the volume of a sphere of that radius (*Figure 5m–n*, *Video 10*). The nearest neighbor distance (NN-distance) was used instead when comparing subsets of cell centers to limit boundary effects (*Figure 5—figure supplement 1a,b*). Both estimates were calculated using *k*-d-trees as implemented in the SciPy library.

For studying cell shape changes upon delamination, single neuroblasts delaminating from the ventromedial aspect of the otic vesicle were chosen for automatic segmentation (n = 5; *Figure 1e–g'*, *Video 2*). Additionally to the reconstructed cell center that defines the lineage of the cell, two more cell centers were added to each cell at each time step to serve as seed points for segmentation. Automatic segmentation of these cells was then generated by the GSUBSURF method (*Mikula et al., 2011*), which is an image segmentation method based on solving a level set partial differential equation of the form

$$\partial_t u = w_a \nabla g \cdot \nabla u + w_c g |\nabla u| \nabla \cdot \left( \frac{\nabla u}{|\nabla u|} \right),$$

where $g$ is an edge detector function depending on the intensity function of the segmented grayscale image and $w_a$ and $w_c$ are positive real constants steering strength of advective velocity field and curvature regularization. An evolution process of this form converges to a steady state, and it is stopped once we do not observe a significant change in the function $u$. The steady state of evolving level set function $u$ provides a segmentation result which can be directly displayed in FIJI together with the raw data or a selected isosurface of function $u$ can represent the surface of the cell. The equation is solved numerically on image voxel grid by the semi-implicit finite volume scheme, and the initial condition is constructed so that we have a reasonable initial approximation of the segmented cell (e.g. a union of spheres or ellipsoids centered in the provided cell centers).

To calculate the volume of the otic vesicle a semi-automatic strategy was applied (*Figure 5—figure supplement 1c–e*), because a fully automatic segmentation of the otic vesicle structure is a rather complicated task – it is not trivial to construct an appropriate initial condition and to distinguish the borders of the vesicle from the borders of the surrounding cells. A set of points (usually containing 100–300 points) lying on the surface of the vesicle was marked manually using FIJI; the inner and the outer surfaces were represented by separate point sets. Then, we used a method for the reconstruction of 3D objects from point clouds based on Lagrangian evolution of surfaces in 3D and the corresponding evolution model reads

$$\partial_t F = w_a (-\nabla d \cdot N) N + w_c d \Delta F + v_T$$

Here, we directly evolve the surface represented by the map (parametrization) $F$. The initial condition is an ellipsoid containing the given point cloud in its inside. The function $d$ is the distance function to the point cloud and it is the driving force of the evolution. The vector $N$ is a unit normal to the surface and $\Delta F$ is its mean curvature. The parameters $w_a$ and $w_c$ are positive real constants. The vector field $v_t$ is a tangential vector field specifically designed to control the quality of the surface discretization mesh during the evolution. The evolving surface is approximated by a triangulated surface. That means that at the end of the evolution, we directly obtain a triangular representation of the otic vesicle surfaces that can be easily used to compute the volume of the vesicle. As in the case of cell segmentation, the evolution model is discretized semi-implicitly in time and by a finite volume method in space.

## Acknowledgements

We are grateful to T Pujol for his valuable help in SPIM imaging and Zeiss for use of the Lightsheet Z.1 microscope. We thank members of Peyriéras and Pujades laboratories for insightful discussions and technical help, especially D Fabreges and G Recher. We acknowledge P Daniel for help in experiments related to otic vesicle point cloud segmentation. We thank V Lecaudey, A Nechiporuk and B Riley for kindly providing transgenic fish lines, and S Fraser for his critical reading of the manuscript and valuable comments. This work was supported from Spanish Ministry of Economy and Competitiveness (MINECO) by Grant BFU2012-31994 to CP, Unidad de Excelencia María de Maetzu 2015–19 MDM-2014–0370 to DCEXS-UPF, Centro de Excelencia Severo Ochoa 2013–17, SEV-2012–0208 and Swiss National Science Foundation (SINERGIA CRSII3 141918) to CRG. BioEmergences services were funded by ANR-10-INBS-04 and ANR-11-EQPX-0029. SD and AZ were recipients of predoctoral FI-fellowships from AGAUR (Generalitat de Catalunya). CP is recipient of an ICREA Academia award (Generalitat de Catalunya).

## Additional information

### Funding

| Funder | Grant reference number | Author |
| --- | --- | --- |
| Unidad de Excelencia María de Maetzu | 2015-19 MDM-2014-0370 to DCEXS-UPF | Sylvia Dyballa Andrea Zecca Cristina Pujades |
| Becas de la Generalitat de Catalunya | Predoctoral FI-fellowship | Sylvia Dyballa Andrea Zecca |
| Agence Nationale de la Recherche | ANR-11-EQPX-0029 | Thierry Savy Nadine Peyriéras |
| Agence Nationale de la Recherche | ANR-10-INBS-04 | Thierry Savy Nadine Peyriéras |
| Schweizerischer Nationalfonds zur Förderung der Wissenschaftlichen Forschung | SINERGIA CRSII3 141918 | Philipp Germann |
| Centro de Excelencia Severo Ochoa | 2013-17 SEV-2012-0208 to CRG | Philipp Germann |
| Ministerio de Economía y Competitividad | BFU2012-31994 | Cristina Pujades |
| Institució Catalana de Recerca i Estudis Avançats | ICREA Academia Award | Cristina Pujades |

The funders had no role in study design, data collection and interpretation, or the decision to submit the work for publication.

### Author contributions

SD, Conceptualization, Data curation, Formal analysis, Validation, Investigation, Visualization, Writing—original draft, Writing—review and editing, Designed all experiments, Carried out all aspects of experiments, Collected and analyzed the data, Assisted with preparing the manuscript, Wrote the

manuscript; TS, Resources, Software, Developed the algorithms in the BioEmergences platform, Assisted in data curation; PG, Software, Formal analysis, Visualization, Provided the Python-based framework for quantitative analysis; KM, Supervision, Supervised the segmentation studies in single delaminating cells and the 3D-point-cloud segmentation; MR, Software, Formal analysis, Assisted in performing the 3D-point-cloud segmentation; RŠ, Software, Formal analysis, Assisted in segmenting single delaminating cells; AZ, Investigation, Assisted in imaging data acquisition; NP, Software, Writing—original draft, Project administration, Provided the BioEmergences platform, Assisted in data analysis; CP, Conceptualization, Formal analysis, Supervision, Funding acquisition, Investigation, Visualization, Writing—original draft, Project administration, Writing—review and editing, Designed and analyzed the experiments, Supervised the project, Wrote the manuscript

### Author ORCIDs
Philipp Germann, http://orcid.org/0000-0002-2057-4883
Cristina Pujades, http://orcid.org/0000-0001-6423-7451

### Ethics

Animal experimentation: This study was performed in strict accordance with the European Regulations. Zebrafish embryos were obtained by mating of adult fish using standard methods. All fish strains were maintained individually as inbred lines. All protocols used have been approved by the Institutional Animal Care and Use Ethic Committee (PRBB-IACUEC), and implemented according to national and European regulations. All experiments were carried out in accordance with the principles of the 3Rs. All our experiments were carried out using the CPC16-008/9125 protocol approved by the Generalitat of Catalonia.

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
