## [Decision Letter]

[Editors’ note: a previous version of this study was rejected after peer review, but the authors submitted for reconsideration. The decision letter after the last round of review follows.]

Thank you for resubmitting your work entitled "Distribution of neurosensory progenitor pools during inner ear morphogenesis unveiled by cell lineage reconstruction" for consideration at *eLife*. Your article has been favorably evaluated by K VijayRaghavan (Senior editor), and three reviewers, one of whom is a member of our Board of Reviewing Editors.

Although all three reviewers are positive about the work and agree that it represents a technical tour de force, the issue of generalising from the low sample sizes has resulted in considerable discussion between the reviewers, Reviewing Editor and Senior Editor. Overall, we are supportive of the work, and so would like to give you the opportunity to address our remaining concerns.

Specifically, the following points should be addressed:

1) The authors need to be much more explicit about stating the sample sizes in the text. Numbers of embryos and numbers of ears, and how these relate to the datasets presented in the tables, should be stated clearly throughout the text.

2) The partial datasets presented in the tables and the numbers of ears/embryos analysed should be supported, where possible, by presentation of the additional data in supplementary form. The reviewers are hopeful that these partial datasets can be used more effectively to lend support to the main findings.

3) The authors should acknowledge and discuss the limitations of their work and the risks of generalising from small sample sizes.

Further detail is given in the full reviews, which are appended below.

*Reviewer #1:*

As before, I am generally supportive of this work, in principle. The writing is improved and all of my minor concerns have been addressed. However, the revised manuscript still does not address my only major concern – e.g. that most data reflect detailed analysis of only single embryos. The tables provided in the revision indicate that multiple embryos were examined for each set of experiments, but analysis of the "extra" embryos does not appear to be shown in any figures, and the bulk of data presented in the main display figures does not include more than one embryo. For example, a single embryo was used for Figure 2 and Video 3, Video 4 and Video 5. It is also possible that this same embryo was used in Figure 3, although this is ambiguous. Similarly, it is possible that only one control embryo and one neurog1 morphant were used for the bulk of data in Figure 4 and Figure 5 and Video 6–Video 10. Information about which embryos were analyzed should be explicitly stated in the figure legends. These figures do provide tremendous detail about a few select embryos, but I am still concerned about whether the patterns and trends they display are reproducible. For the "extra" embryos mentioned in Table 1 and Table 2, presumably the data could be presented in supplemental figures even if they are not as extensive of those in the display figures. Additionally, I feel there should greater effort to consider average trends taken from multiple embryos and somehow display such trends in the figures, such as in Figure 5.

*Reviewer #2:*

This is the second revision of this manuscript that I have reviewed previously. The first revision was primarily a rewrite, with no additional data. In this revision, the authors have added some samples of normal and Neurog1 morphants to quantify the total number of cells in the otic vesicle. A few additional embryos were added to analyze delamination kinetics, location of delamination (using unpublished data provide to the reviewers) and the pattern of hair cell differentiation. I also think this version does a better job of describing how the tracing of single cells through the stages of neurogenesis, gangliogenesis and hair cell formation, adds to the current state of knowledge about inner ear morphogenesis in the zebrafish. The movies are beautiful, the image analysis is sophisticated, and the insights gained primarily concern prepatterning of the neural progenitors based on location and timing of delamination. Additionally, by tracing progenitor proliferation in the neurog1 morphants, the authors ask whether or not the absence of neuroblast delamination could alter the proliferation or behavior of the cells that are thus remaining in the otocyst. This gives rise to otocysts that are larger, but no significant changes in cell behavior. The resulting increase in sensory cells does not, apparently, come from a change in proliferation of the progenitors, but is most likely a fate switch.

*Reviewer #3:*

This manuscript is improved over the previous version and the authors have dealt with many of the referees' comments. It is clear that the work represents considerable effort and will be of use and importance as a reference study for those working on the zebrafish ear. There are interesting observations concerning the place and timing of the emergence of otic neuroblasts from the zebrafish otic epithelium. This is the first time that such a detailed imaging study has been performed on the live embryo to address the process of otic neurogenesis.

I still find that the quantification is not entirely clear. The authors state in their rebuttal that 'our approach provides information about the scale of the system, as stated throughout the manuscript, not in absolute cell numbers that as we know they may differ among individuals'. I find this confusing. What do the authors mean by a difference between scale and absolute cell numbers? The data show individual nuclei and therefore they do appear to provide very precise information about absolute cell numbers, but only in a few individuals, and thus is it hard to generalise these findings.

Information about datasets is now provided in two tables, but it is still very difficult to work out from these exactly how many fish have been used, and how many ears (one or both) from each fish have been imaged. Given the concerns raised from the earlier version of the manuscript, the authors must be up-front in stating exactly how many embryos and ears their analyses are based on. This information is hidden, and the descriptions do not always appear to match the tabled information. For example, the legend to Figure 3—figure supplement 1 states that 'embryos' (plural) were used, but in the table, this transgenic and time combination refers to a single dataset. Is this one ear from one embryo? Information on numbers of ears and embryos should be provided at the start of the Results section, before a description of the findings, in addition to statements in the Materials and methods section and in the Figure legends. The relationship between datasets in the tables and numbers of individuals should be explained. This applies both to the wild-type studies and those in the neurog1 morphants.

---

## [Author Response]

*[…] Specifically, the following points should be addressed:*

*1) The authors need to be much more explicit about stating the sample sizes in the text. Numbers of embryos and numbers of ears, and how these relate to the datasets presented in the tables, should be stated clearly throughout the text.*

To improve this point we made the following changes:

1) We now clearly state at the beginning of the Results section that: “We addressed these questions by exploring in depth and detail a selected number of embryos and support our findings with less extensive analyses of additional embryos, which we provide in supplementary form.”

2) We explain in the Materials and methods (section “Embryo mounting and SPIM imaging”) and in Table 1 and Table 2 legends that each dataset corresponds to the imaging of a distinct embryo inner ear except for 140402aX in which both ears were imaged. We included that we have performed deep and detailed analyses in few datasets and supported the main findings with partial analyses of additional datasets mainly included in supplementary figures.

3) We improved Table 1 and Table 2, explaining the ID datasets characteristics (Table 1) and for what questions they were used (Table 2), and indicating in which figures they were employed (Table 2). We organized the datasets in both tables by order of appearance within the manuscript. We think Table 2 might be a bit repetitive with the information provided in figure legends, but we consider it can be helpful and would like to keep it.

4) We named the ID dataset used in every figure legend, and when a supplementary figure including additional analyses applies (to the presented finding) it is clearly indicated (eg: ID Dataset: 140210aX; see Figure 1—figure supplement 2 for additional analyses).

5) We mention, throughout the text, when not disturbing, the ID Dataset used in that experiment.

*2) The partial datasets presented in the tables and the numbers of ears/embryos analysed should be supported, where possible, by presentation of the additional data in supplementary form. The reviewers are hopeful that these partial datasets can be used more effectively to lend support to the main findings.*

We have made an effort to present better the data obtained from the additional datasets in supplementary figures, and always – when possible – we displayed the data from multiple embryos together to show the average trend. We have included the following:

1) A new panel in Figure 1—figure supplement 2 with the comparative analysis of four ears from three datasets (140210aX, 140225aX, 140306aX; Figure 1—figure supplement 2). This supports the observation made in Figure 1, about the expansion of the delamination territory towards posterior. Accordingly, description of these results can be found in p5.

2) A new supplementary figure (Figure 2—figure supplement 1) displaying the analysis performed with another dataset (140423aX), to support that place and time of neuroblast delamination prefigure their position within the SAG.

3) Two new panels in Figure 4—figure supplement 1 showing that the order of hair cell differentiation and their relative position within the otic epithelium follow the same pattern in distinct embryos (in three ears from two datasets: 140402aX; 140326aX; Figure 4—figure supplement 1).

4) Panel C in Figure 4—figure supplement 1 displays as well the PM hair cell progenitor map obtained with this dataset (ID Dataset: 140326aX). This supports our conclusion that the position of hair cell progenitors foreshadows the organization of differentiated cells within the sensory patch (Figure 4).

5) A new panel in Figure 5—figure supplement 1, displaying a plot with the NN-distances of ventral vs. dorsal or ventromedial vs. dorsolateral territories in four inner ears (ID Datasets: 140507aX, 140519aX, 140430aX, 140426aX; Figure 5—figure supplement 1). This allows to see a common trend: cells within the ventral domain are more compacted that in dorsal territories.

*3) The authors should acknowledge and discuss the limitations of their work and the risks of generalising from small sample sizes.*

We have largely discussed the limitations of our approach at the beginning of the Discussion section, in order to make the reader aware of the sample size.

In addition we have rephrased and properly explained what we meant with 'our approach provides information about the scale of the system. Our point is that these analyses allow us to seize the proportion of the system and to help us understand how it behaves, in terms of proportions rather than in absolute numbers.